# Intervention planning for the REDUCE maintenance intervention: a digital intervention to reduce reulceration risk among patients with a history of diabetic foot ulcers

Kate Greenwell,[1] Katy Sivyer,[1] Kavita Vedhara,[2] Lucy Yardley,[1] Frances Game,[3] Trudie Chalder,[4] Gayle Richards,[5] Nikki Drake,[6] Katie Gray,[3] John Weinman,[7] Katherine Bradbury[1]

KG and KS are joint first authors.

For numbered affiliations see end of article.

**Correspondence to**
Dr Kate Greenwell;
reduceproject@soton.ac.uk

## ABSTRACT

**Objectives** To develop a comprehensive intervention plan for the REDUCE maintenance intervention to support people who have had diabetic foot ulcers (DFUs) to sustain behaviours that reduce reulceration risk.

**Methods** Theory-based, evidence-based and person-based approaches to intervention development were used. In phase I of intervention planning, evidence was collated from a scoping review of the literature and qualitative interviews with patients who have had DFUs (n=20). This was used to identify the psychosocial needs and challenges of this population and barriers and facilitators to the intervention's target behaviours: regular foot checking, rapid self-referral in the event of changes in foot health, graded and regular physical activity and emotional management. In phase II, this evidence was combined with expert consultation to develop the intervention plan. Brief 'guiding principles' for shaping intervention development were created. 'Behavioural analysis' and 'logic modelling' were used to map intervention content onto behaviour change theory to comprehensively describe the intervention and its hypothesised mechanisms.

**Results** Key challenges to the intervention's target behaviours included patients' uncertainty regarding when to self-refer, physical limitations affecting foot checking and physical activity and, for some, difficulties managing negative emotions. Important considerations for the intervention design included a need to increase patients' confidence in making a self-referral and in using the maintenance intervention and a need to acknowledge that some intervention content might be relevant to only some patients (emotional management, physical activity). The behavioural analysis identified the following processes hypothesised to facilitate long-term behaviour maintenance including increasing patients' skills, self-efficacy, knowledge, positive outcome expectancies, sense of personal control, social support and physical opportunity.

**Conclusions** This research provides a transparent description of the intervention planning for the REDUCE maintenance intervention. It provides insights into potential barriers and facilitators to the target behaviours and

### Strengths and limitations of this study

► This research will inform the development of a novel intervention to support the prevention and management of diabetic foot ulcers (DFUs) and is in keeping with recent NICE research priorities for the diabetic foot.

► The integration of theory-based, evidence-based and person-based approaches provided complementary insights into how an intervention could be designed to maximise its acceptability, feasibility and potential effectiveness.

► The REDUCE maintenance intervention plan is comprehensively described and the intervention's potential mechanisms of actions made explicit, thereby increasing transparency and facilitating application of this intervention planning methodology by other intervention developers.

► Although the qualitative sample was representative of patients with a DFU (who tend to be older and may therefore be retired), few younger and employed people were recruited, so their views remain less well understood.

► Although the rapid scoping review allowed scientific evidence to be quickly incorporated into the intervention plan at an early stage, it was not systematic, so it is possible that some literature may have been missed.

potentially useful behaviour change techniques to use in clinical practice.

## BACKGROUND

Foot ulceration is a common, chronic and costly complication of diabetes.[1–3] Healing is slow and recurrence is common, with approximately 40% of patients reulcerating within 12 months.[4–6] The physical and emotional burden of ulceration is considerable; 20%

of ulcers result in amputation and 32% of patients are depressed, which is associated with a threefold greater risk of mortality.[2 7] Although diabetic foot care has been deemed a priority,[2] treatments to prevent ulceration are based largely on expert opinion and small, underpowered, studies.[2 8] Systematic reviews have found no evidence that education alone improves clinical outcomes.[9–12] However, research suggests that psychosocial and behavioural factors may play a central role in healing and prevention.[13]

Evidence suggests an association between longer delays in help seeking and increased ulcer severity, highlighting the importance of regular foot-checking and rapid self-referral.[14] Although physical activity is generally encouraged in diabetes to promote glycaemic control and reduce cardiovascular risk, there is a common assumption that greater physical activity may increase ulceration risk in people at risk of diabetic foot ulcers (DFUs). However, research suggests that moderate, regular activity may decrease risk, or at worst, be unrelated to risk.[15 16] Emotional management may also play a role. Following a DFU, people may experience difficult emotions, including depression, blame and guilt.[17] Depression has been associated with greater ulcer incidence and recurrence and a slower rate of ulcer healing.[18–20] NICE have consequently recommended the development of new interventions targeting such factors.[2]

'REDUCE', a novel complex cognitive behavioural intervention,[21] was developed to reduce reulceration risk and promote healing by modifying associated psychological and behavioural factors.[22] These factors include non-adherence to recommended foot care procedures (eg, foot checking), delayed help-seeking for changes in foot health, low or irregular levels of physical activity and difficulties in managing negative emotions.

REDUCE consists of two phases: an initiation phase of 8-weekly sessions with a nurse or podiatrist to start psychological and behavioural change and a maintenance phase involving two additional sessions held 1 and 3 months later to help sustain these changes. A full description of the intervention can be found in Vedhara *et al.*[22] A feasibility study found REDUCE to be acceptable and feasible for patients and preliminary descriptive findings suggested that patients experienced changes in many of the psychological and behavioural factors targeted by the intervention.[22] However, long-term maintenance of these changes may be more effective if the intervention were available indefinitely and when patients require it. Low-intensity interventions delivered by websites, smartphones or a booklet provide a low-cost solution. This paper describes the planning process for an intervention that will replace the face-to-face maintenance sessions of the original intervention.

The key objective of the REDUCE maintenance intervention will be to provide support to people who have had DFUs to increase their ulcer-free survival with limbs intact (ie, the length of time a patient is free from ulcers without having had an amputation). In keeping with recent NICE research priorities, this will be done through behaviour change and emotional management. It will support people to maintain four behaviours targeted in the initiation phase: regular foot checking, rapid self-referral in the event of changes in foot health, graded and regular physical activity and emotional management.

Published descriptions of complex interventions and their development process are often inadequate, providing readers with little understanding of what the intervention contains, how decisions regarding its development were made and how the intervention is hypothesised to work.[12 23–25] This paper presents the full intervention planning process for the REDUCE maintenance intervention as an example of intervention planning methodology and to increase transparency regarding the intervention's content and hypothesised mechanisms of action. This intervention plan will subsequently inform the development of the REDUCE maintenance intervention.

## METHODS AND RESULTS
### Intervention planning methodology
The intervention planning used theory-based, evidence-based and person-based approaches.[21 26–28] The person-based approach recommends grounding intervention development in an in-depth understanding of the patient and their psychosocial context, gained through qualitative research.[26] Intervention planning included two phases: collating and analysing evidence and creating the intervention plan. Phase I includes two elements: a qualitative and quantitative scoping review and a qualitative interview study. Phase II includes three elements: (1) creating guiding principles; (2) behavioural analysis and (3) logic modelling.

In phase I, a rapid scoping review of qualitative and quantitative literature was used to examine the behavioural and psychosocial needs, issues and challenges of people who have had DFUs. This knowledge was combined with insights gained from a qualitative interview study that explored patients' perspectives on key content and design features for the maintenance intervention. These two studies are both person-based and evidence-based approaches as they aim to develop an in-depth understanding of the patient's perspective (person-based approach), while identifying, summarising and incorporating the evidence-base on the barriers and facilitators to the target behaviours (evidence-based approach). The findings of these two studies were given equal weight when creating the intervention plan.

We also consulted with experts in DFUs, behaviour change and intervention development who belonged to our multidisciplinary project team using regular teleconferences to discuss and gain feedback on drafts of the intervention plan. This team included one diabetologist, two diabetes specialist podiatrists, one diabetes specialist nurse, one cognitive behavioural psychotherapist, five health psychologists and one research psychologist specialising in health. From this, additional barriers

and facilitators were identified and suggestions or refinements to intervention content were made.

In line with a person-based approach,[26] all sources of evidence (ie, scoping review, qualitative study results, expert opinion) were brought together in phase II to create 'guiding principles' that outline the intervention design objectives and key intervention features. Theory-based 'behavioural analysis' and 'logic modelling'[25 28 29] were used to map the evidence and intervention content onto behaviour change theory to comprehensively describe the intervention and its potential mechanisms of action.

## Collating and analysing evidence
### Qualitative and quantitative scoping review
#### Purpose
To review evidence examining the behavioural and psychosocial needs, issues and challenges of people who have had DFUs.

#### Methods
A rapid scoping review of the qualitative and quantitative literature exploring patients' and health professionals' views and experiences of DFUs and their management was undertaken. This was done to ensure that the initial intervention plan was informed by existing evidence from an early stage. A search was undertaken in Web of Science (covering 1970–2017) to ensure coverage of a range of multidisciplinary journals, easily enabling rapid review. This search combined the following terms ('diabetic foot ulcer') AND ('physical activity' OR exercise), ('self-referral' OR 'help seeking'), (check AND (foot OR feet)) and ('emotional management' OR 'mood management'). It incorporated any published research that included patients who had previously had a DFU. Findings regarding beliefs around foot care were excluded if they were only relevant to foot care behaviours not targeted in the REDUCE maintenance intervention (eg, barriers to adherence to prescription footwear). Articles with a biological focus were excluded. Additional literature was identified through expert consultation and article reference lists. Data were extracted on research design, sample size, participants and key findings. Using thematic analysis, the key findings were organised into themes relating to the psychosocial and behavioural issues, needs or challenges to be considered during intervention development.

#### Results
The review identified seven articles and highlighted six themes relating to people's beliefs around DFUs and the target behaviours, challenges people face when engaging in the target behaviours, difficult emotions people may experience following a DFU and concerns about digital interventions (table 1).

### Qualitative interviews
#### Purpose
To explore the acceptability and feasibility of initial ideas regarding the content and delivery of the maintenance intervention from the perspective of people who have had DFUs and to identify potential barriers and facilitators to its target behaviours.

#### Methods
A total of 250 adult (aged 18+ years) patients with diabetes who had previously had a DFU were contacted by letter by their local NHS podiatry service. Participants were excluded if they had a DFU in the previous 2 weeks. Sixty-six patients (26%) expressed interest in the study, 53 of whom (21% of original mail-out) were eligible to participate. Eligible respondents were purposively sampled to represent a diverse set of ages (range: 45–91 years), genders and internet use (table 2). Twenty participants took part in a single semistructured interview.

Interviews explored participants' views of the target behaviours and potential intervention features, including foot checking reminders, facilities for note-taking, personalised advice about when to self-refer, advice on pacing physical activity, goal setting, provision of free pedometers and emotional management techniques. Interviews also explored participants' views on possible modes of intervention delivery, including booklet, website, computer tablet and smartphones and the value of additional health professional input. Ideas for potential content, intervention features and delivery modes were shown on prompt cards. Ideas for intervention features (eg, pedometers) were chosen based on the multidisciplinary team's knowledge of the evidence for the acceptability and effectiveness of these features for changing the target behaviours. Participants were shown an example of an existing diabetes intervention[30] to demonstrate what a website intervention could look like. Interviews were piloted with two people who have had DFUs. See online supplementary appendix 1 for the interview schedule and prompt cards.

Interviews were carried out by KG and KS and took place at participants' homes (n=18) or the university (n=2). Participants were reimbursed for travel and given a £10 voucher. All interviews were recorded and transcribed. KG and KS used thematic analysis to identify potential barriers and facilitators to engaging with the target behaviours and positive and negative perceptions of the potential intervention features and delivery modes.

#### Results
The key findings are outlined below. Example quotes are in table 3.

*Regular foot checking*: Generally, participants perceived foot checking as acceptable and important for preventing DFUs. Many found foot checking easy to do and already checked their feet regularly. However, many participants reported physical limitations (eg, limited mobility) and other physical barriers (eg, wearing casts or bandages) that restricted foot checking. While some people found it easy to spot changes in foot health, others reported difficulties knowing what to look for and in judging whether any changes were problematic. A few described how it

**Table 1**  Key themes identified from the rapid scoping review of the psychosocial and behavioural issues, needs and challenges of people who have had DFUs

| Key themes | Detail from the literature |
|---|---|
| Lack of confidence in foot checking[17 31] | ► Some patients were uncertain about what a DFU was or looked like, what signs of DFUs to look out for and when the DFU was serious enough to seek help from a health professional. Such uncertainties may lead to delays in seeking help. |
| Feelings of lack of control in preventing DFUs[17 31] | ► Some patients felt they had little or no control in preventing further DFUs, as DFUs still occurred even when they were engaging in foot care behaviours.<br>► Some patients believed that they were unable to prevent DFUs. |
| Difficult emotions following a DFU[17 32–35] | ► Some patients were fearful or worried about developing further DFUs, losing limbs through amputation and the impact a DFU reoccurrence might have on their lives.<br>► Some patients felt down or had low self-esteem because of how the DFUs had negatively affected their everyday lives (eg, loss of independence, inability to work and provide for the family, lifestyle changes).<br>► Some patients felt a sense of hopelessness, anger and frustration when DFUs developed despite their attempts to engage in foot care behaviours.<br>► Some patients felt self-blame or guilt for not paying enough attention to their feet, not controlling their diabetes well, not following foot care advice or not engaging in foot care behaviours, especially in the event of reoccurrence.<br>► Some patients experienced social isolation (eg, from restricted mobility, lack of employment) or felt a burden to others because they were dependent on them for daily activities (eg, cooking and driving).<br>► Some patients found it difficult to share their experiences of a DFU with friends and family.<br>► Some podiatrists acknowledged the emotional impact of DFUs on their patients, specifically the presence of anger, depression, anxiety and frustration. |
| Maintaining behaviours long term may be challenging[17] | ► Some patients were not confident that they could maintain foot care behaviours in the long term, with engagement likely to decrease over time.<br>► Some patients were impatient to resume the physical activities they stopped when they had an active DFU, leading them to do too much activity and risk getting another DFU. |
| Physical limitations impeding foot checking[35 40] | ► Some patients and podiatrists reported physical limitations that prevented patients from engaging in foot care behaviours, including joint mobility problems, neuropathy and visual impairment. |
| Concerns over using digital interventions[33] | ► Some patients felt they did not have the necessary computer skills for internet or computer-based interventions. |

DFU, diabetic foot ulcer.

is easy to become lax over time, forgetting to check feet regularly or not thoroughly checking. Participants identified several facilitators to foot checking, including using a mirror to check feet, getting someone else to check and integrating foot checking into everyday routine (eg, when putting on socks).

When discussing the planned intervention features (eg, foot checking reminders, facilities for note-taking), some people believed it would be useful to set up regular email foot checking reminders because it is easy to forget. Others felt reminders could be irritating or were unnecessary, as they, or their podiatrist, already regularly checked their feet. Generally, people thought it would be helpful to be able to make a note of any changes in their foot health to track changes in foot health over time. A few people felt this was unnecessary because they already checked their feet regularly and knew what to look for or believed it would be difficult to remember to note down changes.

*Rapid self-referral in the event of changes in foot health*: Most participants were positive about self-referral, viewing it as important. However, many people found it difficult to contact their DFU team. Long waiting times left some participants worried about how their foot health might decline in the meantime, which led one person to treat their feet themselves, instead of self-referring. In contrast, some participants reported the opposite and found it easy to get an appointment with their DFU team. A few participants were unsure which health professional to contact when reporting DFUs (eg, podiatrist, diabetes nurse, GP). Some expressed concerns about looking foolish or wasting health professionals' time when self-referring for changes in foot health that turned out to be normal. One person had trouble with getting her concerns taken seriously and a few people worried about being a burden to health professionals. Some participants wanted reassurance from health professionals that it was right to have sought help.

**Table 2** Demographics of patients taking part in the qualitative interviews

| Sample characteristics | Statistics |
|---|---|
| Basic demographics | Mean (SD) |
| Age | 68.30 (11.54) |
| Basic demographics | N (%) |
| Male | 11 (55) |
| Marital status | |
| Married | 7 (35) |
| Single | 6 (30) |
| Widowed | 4 (20) |
| Divorced | 3 (15) |
| Employment status | |
| Retired | 15 (75) |
| Redundant due to illness | 3 (15) |
| Housewife/husband | 1 (5) |
| Full-time employed | 1 (5) |
| Educational status | |
| Secondary school | 10 (50) |
| College/Sixth Form/Professional Qualification | 7 (35) |
| Undergraduate | 3 (15) |
| DFU history | Mean (SD) |
| Years since first DFU (approx.) | 6.81 (7.96) |
| Number of DFUs (approx.) | 4.18 (3.86) |
| Months since last DFU (approx.) | 14.65 (11.26) |
| Duration of last DFU in days (approx.) | 298 (400.82) |
| Internet use | N (%) |
| Access to internet at home | 15 (75) |
| Access to internet on tablet | 7 (35) |
| Access to internet on phone | 3 (15) |
| Frequency of access | |
| Never | 3 (15) |
| Less than once a month | 3 (15) |
| Once a week | 1 (5) |
| A few times a week | 2 (10) |
| Once a day | 3 (15) |
| Several times a day | 8 (40) |

DFU, diabetic foot ulcer.

*Graded and regular physical activity*: Most participants were positive about physical activity, stating that they would like to or were already doing it. People generally viewed physical activity as important for general health and diabetes management. However, many participants reported physical limitations (eg, pain, fatigue) or diabetic complications (eg, neuropathy, residual damage to feet from previous DFUs) that made it difficult to be active. Participants reported that it was important to find the right activity to overcome their physical limitations, suggesting activities that did not put pressure on their feet, such as seated exercises. Some were concerned that physical activity might cause another DFU or exacerbate other health conditions.

Some participants stated that it could be difficult to maintain physical activity. A few mentioned that integrating physical activity into their daily routine (eg, getting off the bus one stop early) and positive encouragement helped. Participants viewed self-monitoring, goal setting and pedometers as helpful for maintaining motivation. However, some people disliked the idea of being 'spied on' or told what to do, expressed doubts about the accuracy of pedometers or were unsure whether they would use them.

*Emotional management*: Over half of participants viewed emotional management positively and reported experiencing low mood, frustration, anger and stress either during or after a DFU. Others had not experienced such emotions relating to their DFUs and viewed emotional management as irrelevant. A few people viewed emotional management negatively due to previous negative experiences. For example, some had experienced unhelpful reactions from doctors when discussing emotions, disliked talking about their feelings in counselling or had received unhelpful information about emotional management (eg, being given advice that did not consider their physical limitations). Some expressed a lack of understanding about how the emotional management would help or perceived it as contrary to their personal style of managing emotions (ie, ignoring their problems, 'getting on with it').

*Intervention delivery methods*: Most participants were positive about the idea of the intervention being delivered via a booklet. Booklets were perceived as quick and easy to refer to, portable and easily shared or distributed (eg, with relatives or picked up from clinics). However, some participants commented that booklets were easily misplaced or forgotten. Most internet users reacted positively to the idea of a website, mainly because it was easy to access, convenient and had interactive features (eg, quizzes, email reminders). Nonetheless, non-users and a few infrequent internet users expressed concern about their own computer literacy. Some participants disliked reading on a computer screen and a few participants had concerns about security of web interventions. However, when participants were shown the example website, they generally viewed it positively, stating that it looked easy to use. A few participants would have liked to access the intervention using a computer tablet as they already used one or knew people who did. Most viewed delivery using a smartphone negatively because of their limited use of phones or difficulties with using small screens due to poor eyesight (caused by diabetes). A few participants commented that it might be helpful to deliver the intervention through multiple modes (booklet, website, tablet or phone).

Generally, participants were in favour of additional health professional support. However, they interpreted

**Table 3** Key issues arising from our qualitative study and illustrative quotes

| Issue arising from our qualitative study | Participant quotes |
|---|---|
| **Foot checking** | |
| Some participants had physical limitations that make it difficult to check their feet. | 'As you get older you're not so mobile so you can't see right underneath [your foot], so it's a bit of guesswork until you do go…to [the] podiatrist' (P10, Male) |
| Some people found it difficult to know what to look for when foot checking and when to self-refer. | 'Recognising them [DFUs] I think is the hardest part' (P14, Male)<br>'Sometimes…I go [to the podiatrist] and it's not an ulcer…but I can't tell' (P8, Male) |
| A few participants found it difficult to keep up foot checking long-term. | 'You kind of become rather lax about perhaps doing it [foot checking] properly' (P1, Male) |
| There were mixed views on foot checking reminders. | 'I don't think I would need to be reminded. I'm doing it [foot checking] already, really' (P3, Female)<br>'It's nice to have a reminder. Sometimes you get a bit complacent and you think "Oh, I'll do it next time" ' (P10, Male) |
| **Rapid self-referral** | |
| Some participants found it difficult to contact and get an appointment with their DFU team. | 'Sometimes you can't get appointments…By the time you are seeing somebody it's either through [Accident and Emergency], because you've been rushed in 'cause your foot's swollen up and changed colour' (P18, Female) |
| Some participants expressed concerns about self-referring. | 'If you do that [point out changes in foot health] every visit and it's nothing to worry about, you're paranoid, micromanaging. But if you don't mention something you've seen previously, you're complacent and don't care about your health. You can't win' (P18, Female) |
| Some participants found it difficult to know which health professional to contact when reporting DFUs. | 'Who do you contact if you have a problem? Your own doctor? Or the nurse, diabetic nurse? Or the podiatrist?' (P5, Male) |
| **Physical activity** | |
| Some participants have physical limitations that make it difficult to engage in physical activity. | 'I get very breathless. I don't walk much at all. I know I should, but I don't' (P3, Female) |
| Some participants also expressed concerns about physical activity causing another DFU. | 'Even though you might not have an ulcer, even if you go back to minimal activity…you can still get that ulcer come back' (P18, Female) |
| Some participants found it can be difficult to keep up with physical activity over time. | 'It is easy to find something else to do [instead of physical activity]. You've got to be pretty disciplined' (P6, Female) |
| There were mixed views on pedometers. | 'The pedometer is a really good idea though…It's like a game—you want to make sure you can get as many steps in" (P20, Female)<br>'[The pedometer is] almost like being spied on' (P14, Male) |
| **Emotional management** | |
| Emotional management was relevant and valued by some participants, but not everyone. | 'I'm one o' these anxiety merchants, me. I worry for the world…so it'd [emotional management] be very helpful' (P10, Male)<br>'I don't think personally I would have taken it [emotional management] on board at all…it's not gonna make any difference to me…I just think I've got it [DFUs], I've got to put up with it…I don't want to sit on a couch breathing in and out, I want to get on and do something' (P2, Female) |
| **Delivery methods** | |
| Participants were positive about the idea of a website, but there were some concerns about computer literacy. | 'Personally think the website would be far better than the booklet…It's prodding me to do it [use the intervention]…If it's in a leaflet, it just gets left ' (P14, Male, internet user)<br>'I love…anything interactive like that [the quiz in the example website] I think is great…you feel part of it [the intervention], rather than just being dictated to…[the information] tends to sink in better' (P20, Female, internet user)<br>'If I was competent…I would do it on the computer. But I'm not competent' (P8, Male, infrequent internet user) |

Continued

**Table 3** Continued

| Issue arising from our qualitative study | Participant quotes |
|---|---|
| A booklet might be helpful for quick reference and for those who do not use the internet. | 'A booklet is always there, you can always refer to it, you've got something in black and white' (P8, Male) |
| Delivering the intervention via smartphone was less acceptable. | 'Mobile phone—you've got all the problems of the computer, but on a smaller screen…a lot of diabetics [have] got problems with their eyes as well' (P17, Male) |
| Participants liked the idea of additional health professional support, but not for the intended purpose of supporting behaviour maintenance. | 'It'd [additional health professional support] give me the confidence to know that 'well, I am alright with my foot as it is'…because you can get a bit paranoid over it [your foot health]' (P17, Male)<br>'They could give…one-to-one advice on…is there anything else that you could do… better than what I'm doing myself' (P3, Female) |

DFU, diabetic foot ulcer.

this as support to gain reassurance about the status of their foot health and advice on foot care or when to self-refer (which would be covered in the website/booklet), rather than support to raise motivation for engaging with the target behaviours. Very few participants said they might use this support to answer questions about information in the booklet or website.

Explanations of how the evidence from the scoping review and qualitative study informed intervention planning are provided in the next sections on Guiding Principles and Behavioural Analysis.

### Creating the intervention plan
#### Creating guiding principles
##### Purpose
In line with the person-based approach,[26] brief guiding principles are developed and consulted throughout intervention development to ensure that the intervention is underpinned by a coherent focus.

##### Methods
Drawing on the findings from our scoping review and qualitative study, key characteristics of target users and the key behavioural issues, needs and challenges the intervention must address were described. From this, guiding principles were created, which outline the intervention design objectives that will address these key behavioural issues, needs and challenges and the key intervention features designed to achieve these objectives. The multidisciplinary team decided on the key features based on their ability to address the intervention objectives.

##### Results
People who have had DFUs can feel they have little or no control over preventing DFUs, as DFUs can occur even when people are engaging in foot care behaviours. This leaves people feeling hopeless and frustrated.[17] Some people may feel self-blame or guilt for not engaging in foot care behaviours, especially in the event of reoccurrence.[17] Therefore, one design objective was to reduce feelings of hopelessness, frustration, self-blame and guilt following a DFU.

People may be uncertain about the signs of a DFU and when to seek help from a health professional.[31] Our qualitative study highlighted that some people were concerned about looking foolish, being a burden or wasting healthcare professionals' time if changes in their feet turn out to be normal. This may delay help seeking. Therefore, one design objective was to build patients' confidence in making a self-referral.

This population is likely to have physical limitations and/or comorbidities. Our qualitative study highlighted that these challenges may make it difficult for people to engage in foot checking and physical activity. They may also be reluctant to increase activity in case it causes reulceration. Thus, one design objective was to acknowledge that patients may have physical limitations that make it difficult to engage in foot checking and physical activity.

Our scoping review highlighted that people may experience difficult emotions following a DFU.[17 32–35] However, some participants in our qualitative research did not experience such emotions and, therefore, did not perceive emotional management as useful. Therefore, one design objective was to acknowledge that emotional management may not be relevant for all patients.

As the physical activity and emotional management content was not relevant to all patients, these components were made optional, rather than mandatory, to avoid discouraging patients from engaging in the other target behaviours if they do not want to increase physical activity or engage in emotional management.

In our qualitative study, many reacted positively to the idea of a web-based intervention, but some participants expressed concerns about their computer literacy. These concerns were also evident in the literature.[33] Therefore, one design objective was to ensure people feel confident in using the maintenance intervention. We decided to deliver the intervention using a website and provide key information and advice in a booklet for quick reference and for non-internet users. At the preceding initiation phase, health professionals will address concerns and speak favourably of the digital intervention to encourage

**Table 4** The guiding principles for the development of the REDUCE maintenance intervention

| Intervention design objectives | Key features |
|---|---|
| To reduce feelings of hopelessness, frustration, self-blame and guilt following a DFU | ► Emphasise target behaviours that patients can engage in to reduce their chances of getting another DFU, while acknowledging that there are precipitating factors (eg, increased age, neuropathy, foot shape) that are out of their control.<br>► Enhance patients' confidence in the target behaviours (eg, by providing a rationale for the necessity of the target behaviours, scientific evidence that behaviours are effective, patient stories and a quiz on the benefits of the behaviours).<br>► Validate patients' feelings of frustration and hopelessness if a DFU does reoccur and avoid arguments that may be viewed as blaming patients for this reoccurrence.<br>► Provide links to emotional management techniques that can help people to manage difficult emotions. |
| To build patients' confidence in making a self-referral | ► Provide links to foot checking training (eg, by providing information and photographs on what DFUs look like, what signs to look out for and how often feet should be checked with guided practice).<br>► Provide reassurance that self-referral is necessary (eg, through a foot health checklist that provides personalised feedback on whether or not patients should self-refer, based on their symptoms).<br>► Address concerns around looking foolish or wasting the DFU team's time when self-referring (eg, (1) emphasise that the DFU team would rather they were contacted early so they are better able to treat any DFUs, (2) provide patient stories about how other patients overcame feelings of burden). |
| To acknowledge that patients may have physical limitations that make it difficult to engage in foot checking and physical activity | ► Provide guidance on how to check your feet if you have physical limitations, including using a mirror to check the bottom of your feet and asking someone else to check for you.<br>► Make intervention content on physical activity optional.<br>► Provide guidance about a variety of safe and low impact physical activities to enable patients to find an activity that is suitable for them.<br>► Address physical activity concerns all the way through the intervention (ie, in the maintenance intervention and prior initiation phase) (eg, by providing information about the safety of physical activity, patient stories about how other patients overcame these barriers). |
| To acknowledge that emotional management may not be relevant for all patients | ► Make intervention content on emotional management optional.<br>► Emphasise that some people, but not everyone, might experience difficult emotions following a DFU to avoid excluding those who may not relate to this content.<br>► Provide a variety of brief emotional management techniques (eg, cognitive behaviour therapy, mindfulness techniques) to allow each person to find a technique that fits with their own personal style of managing emotions. |
| To ensure patients feel confident in using the maintenance intervention | ► Keep website navigation simple and follow guidelines for maximising website usability.<br>► Health professionals at the prior initiation phase will provide technical support, address self-doubts and speak favourably of the digital intervention to encourage use.<br>► Encourage friends and family to assist people with website use, if appropriate.<br>► Provide a booklet for quick reference and for those who do not have access to the internet. |

DFU, diabetic foot ulcer.

use. Table 4 details the REDUCE maintenance intervention guiding principles.

### Behavioural analysis
*Purpose*
To use behaviour change theory to systematically describe the maintenance intervention content, identify potential determinants of behaviour (ie, what needs to change for a behaviour to occur) and map it onto the evidence derived from our scoping review, our qualitative study and expert consultation.

*Methods*
Behavioural analysis involves comprehensively mapping out the elements of an intervention, linking the evidence-base to behaviour change theory and the intervention

components. Providing a clear description of the intervention is essential for replication in research and practice, data extraction in systematic reviews and process evaluation planning.[21 24 25] The Behaviour Change Wheel (BCW[36 37]) and Behaviour Change Techniques Taxonomy (BCTv1[38]) were developed to standardise the classification and description of complex interventions and help identify an intervention's 'active ingredients' and behavioural determinants. Such standardisation provides a common language to avoid any confusion that may occur when different terminology are used for the same intervention technique or different techniques are referred to using the same terminology.[39] The BCW draws on the COM-B model, which argues that behaviour is influenced by an individual's Capability, Opportunity and Motivation to change behaviour.[37]

In addition to the four target behaviours identified from the outset, the behavioural analysis also identified one subsidiary behaviour (engaging with the digital maintenance intervention) that is necessary to enact these target behaviours. Barriers and facilitators for each behaviour were identified from the primary qualitative research, scoping review and expert opinion from the multidisciplinary project team. Intervention components that addressed each barrier and facilitator were selected. These components are reported using patient-centred, autonomy-supportive language to emphasise the importance of delivering these components in a way that will enhance intrinsic motivation and ensure a positive intervention experience.[26] The intervention components were coded using the BCTv1 and mapped onto the BCW to identify their corresponding intervention function (ways an intervention can change behaviour, eg, 'education') and target construct (what needs to change for the behaviour to occur, eg, 'psychological capability'). The BCTv1 and BCW were then examined to check for potentially useful additional intervention functions, target constructs or behaviour change techniques.

### Results

The behavioural analysis is presented in online supplementary appendix 2. The maintenance intervention will target all six behavioural sources included in the BCW (physical and psychological capability, reflective and automatic motivation and physical and social opportunity) and employ six different BCW intervention functions (education, persuasion, modelling, training, enablement, environmental restructuring) using 18 different BCTs. Intervention components that received a mixed reaction from our qualitative research participants (ie, foot checking reminders, pedometers) were made optional to promote patient autonomy.

Although participants would have liked additional health professional support, the support participants wanted was more clinical in nature (eg, advice about foot health or when to self-refer). As such support would be provided in the website/booklet, this form of health professional support was deemed superfluous.

Therefore, additional health professional support was not included in the intervention plan. One issue that arose from our qualitative study could only be addressed to a limited degree by the maintenance intervention, namely the difficulties people experienced contacting and getting an appointment, with their DFU team. This will be addressed by educating patients about the national guidelines and local procedures for self-referrals and how to communicate the reason for self-referral to their DFU team. However, improving local self-referral pathways or modifying health professionals' behaviour is outside of the scope of this intervention.

### Logic modelling

*Purpose*

To model the hypothesised mechanisms of action of the maintenance intervention (ie, how it is thought to work).[25 28 29]

*Methods*

The logic model draws together findings from the scoping review, qualitative study and behavioural analysis into a testable model that outlines how the different intervention components are hypothesised to impact on subsequent components and ultimately affect outcomes.

*Results*

The logic model (figure 1) can be broken down into three major components.

*Intervention techniques and processes*: The intervention techniques summarise the behaviour change techniques outlined in the behavioural analysis and the seven processes they are hypothesised to affect: skills, self-efficacy, knowledge, positive outcome expectancies, sense of personal control, social support and physical opportunity. These are the psychosocial factors that need to be modified for the intervention's target behaviours to change and were identified through the behavioural analysis.

Each set of intervention techniques is hypothesised to mainly affect one of these processes, which subsequently affect one or more of the intervention's target behaviours. They are organised in order of importance, with more integral processes that were consistently identified as key in the scoping review and qualitative study at the top and less integral processes at the bottom (eg, optional features).

*Purported mediators*: Purported mediators are the target behaviours of the intervention that are hypothesised to directly affect DFUs in the long term. These behaviours are divided into 'core behaviours' that are hypothesised to be most important in determining DFU outcomes (foot checking, rapid self-referral) and 'optional behaviours' that are only relevant for some patients (physical activity, emotional management). These behaviours may impact either directly, as in the case of physical activity, or indirectly, via their effect on the other target behaviours, as is the case in emotional management. Emotional management is hypothesised to have an indirect effect on the

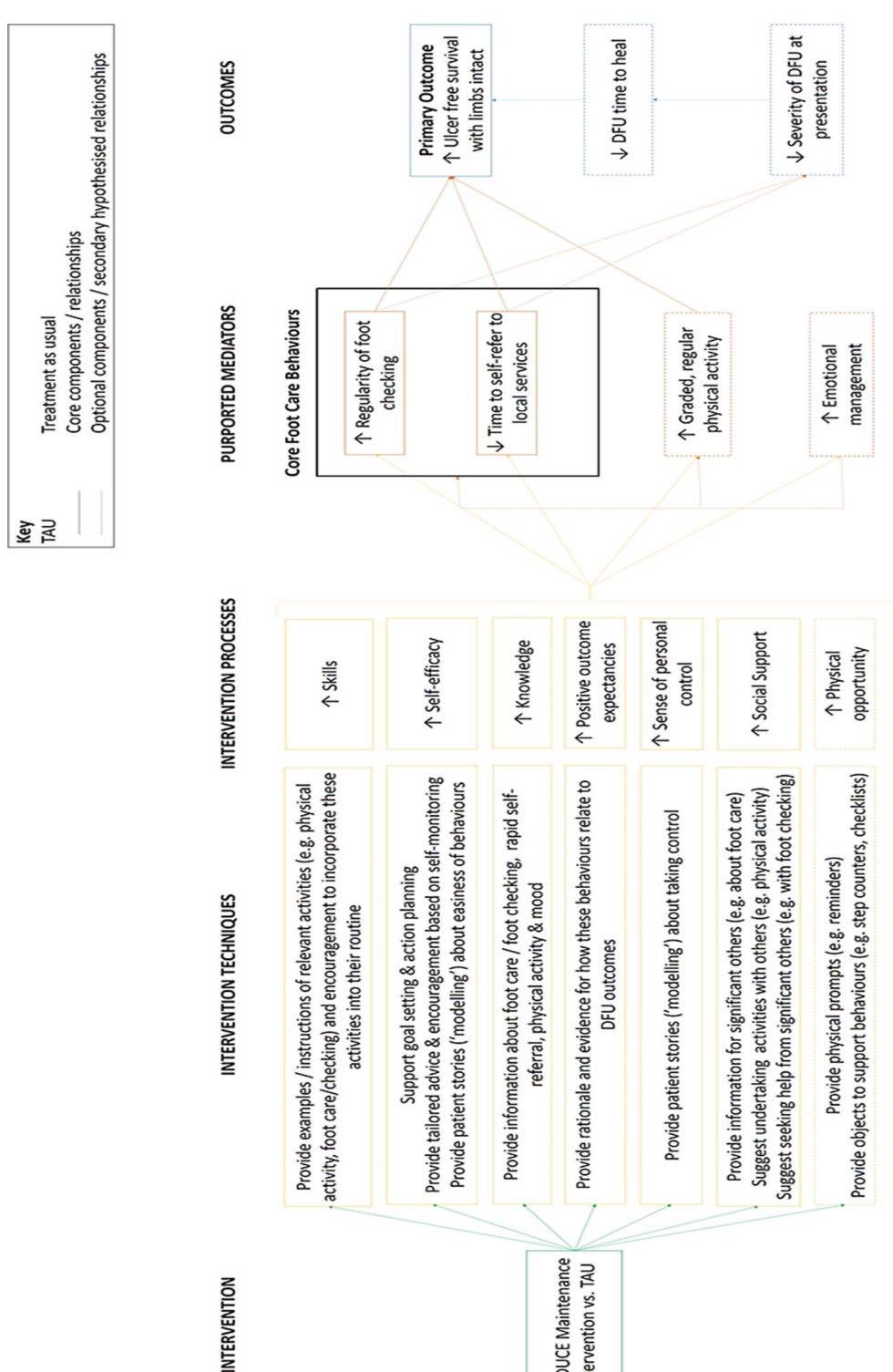

**Figure 1** REDUCE maintenance intervention logic model. DFU, diabetic foot ulcer.

other behaviours due to the negative effects that low mood (or negative thoughts) can have on behavioural engagement.

*Outcomes*: The logic model specifies three outcomes that the intervention is ultimately trying to change, the primary outcome of interest (ulcer-free survival with limbs intact) and two interim outcomes that may be affected by the target behaviours and may, directly or indirectly, affect the primary outcome (severity of DFU at presentation and time taken for DFU healing in the event of a recurrence).

## DISCUSSION

This paper describes the use of theory-based, evidence-based and person-based approaches[28] to developing an

intervention plan for the REDUCE maintenance intervention, an intervention that aims to reduce reulceration risk by supporting patients to maintain behaviour change and emotional management. These different approaches provided complementary insights into how the intervention could be designed to maximise its acceptability, feasibility and effectiveness. For example, the scoping review highlighted that patients experience difficult emotions following DFUs;[17 32–35] however, the qualitative interviews suggested that this was only relevant for some patients, suggesting that this content should be made optional. In line with person-based and evidence-based approaches, our scoping review and qualitative study deepened our understanding of the psychological and behavioural needs of people who have had DFUs and highlighted several barriers and facilitators to the intervention's target behaviours, some of which had been highlighted in the literature (eg, lack of knowledge regarding what to look for when foot checking[17 31]) and some which had received little prior attention (eg, lack of knowledge about when to self-refer). It also highlighted important advantages of, and barriers to, successful use of different intervention delivery methods (eg, lack of confidence in ability to use digital interventions). Our qualitative study updated prior research published over a decade ago that highlighted concerns regarding limited computer access and poor computer skills among people at risk of DFUs.[33] Our guiding principles succinctly summarised the distinctive design objectives and features of the maintenance intervention, while our behavioural analysis and logic modelling comprehensively described the intervention and its potential mechanisms of action.

This is the first paper to use this methodology to provide a comprehensive plan of a DFU intervention. Transparent reporting of the intervention planning process will allow other researchers to easily understand how this methodology could be applied to different intervention contexts and facilitate comparison between different interventions.[12 23–25] The use of primary qualitative research allowed us to understand patients' views on the delivery methods for behaviour change interventions and three behaviours that have received little attention in the DFU literature to date: engaging in rapid self-referral, graded and regular physical activity and emotional management. For example, participants had mixed reactions to some behaviours (ie, physical activity and emotional management) and design features (eg, email reminders), which were subsequently made optional. Participants also reported experiencing difficulties with accessing their DFU team when self-referring. Future research should further explore and address any professional and organisational barriers to self-referral.

The qualitative research used purposive sampling which enabled us to explore the acceptability and feasibility of a digital intervention across a diverse set of people, including those who were frequent and infrequent internet users. Although the sample was representative of the population of people with DFUs (who tend to be older[14] and may therefore be retired), it would be helpful to explore the views of younger and employed people, as they may report different barriers to behaviour change. The rapid scoping review allowed scientific evidence to be quickly incorporated into the intervention plan, but it was not systematic, so it is possible that some literature was missed.

Recent NICE guidelines for the prevention and management of diabetic foot problems[2] identified a need to develop and evaluate new interventions targeting psychological and behavioural factors. Our research has provided a plan for such an intervention as well as identified potential barriers to behaviour change and behaviour change techniques that are likely to be useful within clinical practice. In future work, we intend to use this intervention plan to develop the maintenance intervention and then conduct an effectiveness trial to evaluate the effectiveness and cost-effectiveness of the entire REDUCE intervention, while also examining if the intervention works as hypothesised.

### Author affiliations
[1]Centre for Clinical and Community Applications of Health Psychology, University of Southampton, Southampton, UK
[2]Division of Primary Care, School of Medicine, University of Nottingham, Nottingham, UK
[3]Department of Diabetes and Endocrinology, Derby Teaching Hospitals NHS Foundation Trust, Derby, UK
[4]Department of Psychological Medicine, Institute of Psychiatry, Psychology, and Neuroscience, King's College London, London, UK
[5]Department of Diabetes, Northern Devon Healthcare NHS Trust, Barnstaple, UK
[6]Podiatry Department, Bristol Community Health, Bristol, UK
[7]Institute of Pharmaceutical Science, King's College London, London, UK

**Acknowledgements** The research team acknowledges the support of the National Institute of Health Research Clinical Research Network (NIHR CRN). We are grateful to the members of the REDUCE PPI panel for their valuable input on the study materials and interview schedule and the participants who took part in this research.

**Contributors** All authors designed the study. KGre, KS, KB and LY led the intervention planning, with input from the other coauthors. KGre and KS were responsible for recruitment, carrying out the interviews and analysing the data, with support from KB. KGre and KS jointly drafted the manuscript with initial support from KB and LY. All authors critically reviewed the manuscript, contributing important intellectual content and approved the final manuscript.

**Funding** This article summarises independent research funded by the National Institute for Health Research (NIHR) under its Programme Development Grants Programme (Grant Reference Number RP-DG-0615-10005) and the NIHR Biomedical Research Centre at South London and Maudsley NHS and Kings College London.

**Disclaimer** The views expressed are those of the author(s) and not necessarily those of the NHS, the NIHR or the Department of Health and Social Care.

**Competing interests** KGre, KS, KV, LY, FG, TC, ND, GR and KB had financial support from NIHR for the submitted work; TC had other support from NIHR.

**Patient consent** Not required.

**Ethics approval** North West – Greater Manchester West Research Ethics Committee (17/NW/0024).

**Provenance and peer review** Not commissioned; externally peer reviewed.

**Data sharing statement** No additional data are available.

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
