## [Reviewer comments · BMJ Open]

ARTICLE DETAILS

TITLE (PROVISIONAL)	Intervention planning for the REDUCE maintenance intervention: a digital intervention to reduce re-ulceration risk among patients with a history of diabetic foot ulcers
AUTHORS	Greenwell, Kate; Siwyer, Katy; Vedhara, Kavita; Yardley, Lucy; Game, Frances; Chalder, Trudie; Richards, Gayle; Drake, Nikki; Gray, Katie; Weinman, John; Bradbury, Katherine

VERSION 1 – REVIEW

REVIEWER	Katherine Brown Coventry University, UK
REVIEW RETURNED	03-Nov-2017

GENERAL COMMENTS	Use of the abbreviation MI in the abstract - should be explained. Table 4 provides the guiding principles and links these to the key features that addresses them. The reader has not as yet however, been provided with any of the other critical detail that has led to translation of qualitative analysis, literature review and guiding principles into intervention content - feels like we're jumping ahead here. You could consider moving it further down. I think the behavioural analysis table could be presented in the main paper rather than as an appendix
---

REVIEWER	Dr Lisa Farndon Sheffield Teaching Hospitals NHS Foundation Trust, United Kingdom.
REVIEW RETURNED	13-Nov-2017

GENERAL COMMENTS	This is an interesting paper which offers insight into possible considerations when educating patients about the diabetic foot. Psychosocial factors have often been 'missed out' in previous research on diabetic foot education programmes. I have a few minor revisions. In the abstract section MI needs to be written out in full on first mention and again on Page 21, line 49. Background section, page 4, line 17. I disagree that there is a lack of evidence based treatments. There are lots of treatments for diabetic foot ulceration but the main problem is there is a lack of evaluation for the preventative measures on a large scale(regular screening and surveillance, rapid referral etc) and whether NG 19 NICE guidance is being followed by all NHS podiatry/diabetic foot services. Please can the authors justify this statement or clarify. Page 6 methods sections. Please provide the date range of which
--

	papers were included in the rapid review and justify. In the results of the rapid review please can the authors provide information on how the 7 themes were derived, e.g. framework analysis?. Page 9, methods sections. The second paragraph on how the patients were identified should be moved to the opening paragraph of this section to guide the reader on the process.
--	--

REVIEWER	Peter Lazzarini Queensland University of Technology, Australia
REVIEW RETURNED	28-Nov-2017

GENERAL COMMENTS	BMJ Open – Manuscript Reviewer Report for: Title: Intervention planning for the REDUCE maintenance intervention: a digital intervention to reduce re-ulceration risk among patients with a history of diabetic foot ulcers Article Type: Original Research Reviewer’s Report: Thank you for the opportunity to review this very interesting paper describing the development of a plan to develop an electronic self-care intervention to help patients with a history of DFU stay in remission. This paper has some really valuable and unique findings that very much add to a very important emerging self-care field of diabetic foot research, i.e. engaging patients in care. I think their unique findings need to be published. However, I couldn’t help but feel the authors were also trying to fit too much into one paper. It felt like I was reading a summary of an entire thesis in one paper that perhaps didn’t do the individual mini-studies in the paper justice. I would suggest the authors consider splitting this paper into 2 or 3 papers as the qualitative study could easily have been a paper in its own right and I thought it was a very nice study that got a little lost with everything else going on. If the authors do consider to keep going with one paper as is, may I suggest you consider explaining the original REDUCE program a little better, how this development procedure builds on the original
---

program and that you are intending to then develop a further intervention after this paper from what you have learnt from this paper. Also I'd suggest 'signposting' the 4-5 mini-studies/phases a little better early in the Methods and then perhaps consider writing the paper as a typical research paper, i.e. firstly a methods section outlining the methods of the 4-5 phases of development and then a results section reporting the results of these 4-5 phases; rather than combining all into one big section as it is at present.

In saying all the above, I reiterate I think the authors have done some very nice work here and have some very unique and valuable findings that need to be published, but at the moment these valuable findings get a little lost in the overall presentation of the paper. I think with some revision of the paper in a perhaps more logical fashion and flow, or splitting it into 2-3 papers, then this paper should be published by BMJ Open very soon. I congratulate the authors for taking on this much needed work and I look forward to citing their paper/s in the near future.

My specific comments are listed below:

Abstract/Strengths and Limitations

1. Page 2, Line 6. Please spell out "REDUCE" or briefly explain what it is. Also please consider this for the title as well.
2. Line 13: As per the main paper I think it would also be useful to explain a little better here how the "mini-studies" in the methods fit together into the larger paper perhaps as escalating phases
3. Line 36: As per the main paper I'm not sure you can broadly conclude the "key challenges facing patients" were essentially the four targeted behaviours chosen by the authors prior to the methods. Perhaps just clarify that these challenges emanated from the four targeted behaviours only
4. Line 45: What is MI? I can't remember it being mentioned in the main paper
5. Line 49. Are the "intervention processes outlined" here a

finding of your study or just what the literature suggested should be broadly covered in a logic model framework?
Possibly consider removing this sentence.

6. Page 3, Line 15: This point is great, but I didn't get this in the main paper until the last sentence of the Discussion, perhaps add something like this in the Background of the main paper somewhere.

Background

1. Page 4, Line 17: Suggest cite this statement re: lack of treatments to prevent ulceration as there are papers in References that will support this important statement in your rationale.
2. Line 24: Perhaps be more specific with "such factors" that NICE have recommended, ie what were those factors as this would add nicely to your rationale.
3. Line 29: Suggest explaining 'REDUCE" a little better. At the moment it is just a little confusing as the program title seems to be an acronym that's not spelt out or a title for a more general cognitive intervention program that is now being applied to diabetic foot disease or is it just the name of this specific new cognitive intervention for foot disease that recently been developed? I'm guessing the later but just clarify please
4. Line 42: Did REDUCE produce improved (patient) outcomes or was it just a pilot feasibility study? Please briefly add a sentence or two on what its outcomes were that you are now intending to build on.
5. Line 44: I think I know what you mean by "digital maintenance" but perhaps clarify this a little better, ie you are essentially trying to build on the REDUCE program and also turn it into a web-based electronic design?
6. Line 49: Possibly add a sentence to Segway a little better between these paragraphs to say REDUCE maintenance intervention is what you are going to do in this study. At the moment it sounds like you are still discussing someone else's program and not the one you are just about to describe in your paper.
7. Page 5, Line 17: Would "describes" be a better term than

“presents” to give the reader the impression that this will be a descriptive paper?

Methods and Results

1. Its not my area of expertise; however, I'm not familiar with essentially reporting 4 or 5 small studies consecutively such as this in one large paper like a mini-thesis. I'm wondering if it would be easier for the reader to conceptualise using phases (as you seemed to have used four escalating phases) and report those phases under one methods section and then report the results of those phases in one results section instead of all together in one big section? Also how does this plan build on the original REDUCE program? We don't seem to have any details on the original REDUCE program but it seems like the reader should know what it is.
2. Page 5, Line >31: I'm not sure this introductory-type paragraph set the reader up that well for the multiple studies (or phases) that followed. Perhaps towards the end of this intro very clearly state for the reader the mini-studies/phases that will follow in the methods (even number them) so the reader is not that surprised when they start reading mini-studies/phases as I was initially, eg “This study was undertaken in 4 phases: i) literature review, ii) qualitative interviews, iii)”
3. Line 48: I'm not sure the “Additional barriers and facilitators ... were identified through consultation with our multidisciplinary project team, including clinical and intervention development experts.” was described later on like the other mini-studies/phases were. Consider adding something on how you obtained this “expert opinion”.
4. Page 6, Line 15: Would it be better to use “review” than “collate”?
5. Line 22: Were you looking for health professional's views as well as patients? The results seem to indicate patients only. I would have thought patients only would suffice though, but remove health professionals if that was the case.
6. Line 27: It would be useful to provide some rationale why

	the particular four behaviours were chosen as I assumed this was going to come from the rapid search exploring experiences of managing DFU or DFU history. However, the search only used terms for the four behaviours. Possibly also add this explanation to where they were mentioned in the Background as well.  7. Line 48: I think the authors need to state the seven articles and themes were those applicable to the 4 behaviours they specifically targeted rather than relating this to all patient beliefs around foot care etc. 8. Page 8, Table 1: In the results it said 7 themes were elicited from the rapid lit review but there are only 6 themes in the table 9. Page 9, Line 8: I think this mini-study needs some brief explanation of what the intervention was that was presented to the patient participants. Further, this mini-study doesn't seem to link to the first mini-study that well and seems to beg the question why was study 1 performed when it was always the intention to use a program already developed? Perhaps clarify how these 2 mini-studies fit together a little better. 10. Line >17: Suggest some of the methods are more findings (results) of the methods and to put them in results, e.g. "66 patients expressed interest ... " and Table 2 should probably go under results also. 11. Page 11, Table 3: There are some really nice and unique results in this table and in the following few pages from your qualitative study here. This study could be a paper on its own. Perhaps move the table down under some of the first results text though as I didn't realise there were more results summarised in text 12. Page 12, Line 5: I'm still not sure why any of the 4 targeted behaviours were chosen? I can assume why a few of them were, however, physical activity for example seems an odd one out as limited solid evidence on what type and intensity of physical activity promotes ulcer prevention and also where the "pedometer" discussion came from? Again it would be useful to have some rationale around why/how these 4 targeted behaviours were originally chosen. If it was from the original REDUCE study [14] then this rationale
--	---

	needs to be in the background, but if it was part of this study then it should go in the methods somewhere perhaps? 13. Page 13, Line 35: What were the planned intervention features? It would be useful to have described them before here I think 14. Page 16, Line 17: As per points above this paragraph I think would be better in the Methods introductory paragraph to explain to the reader how all these mini-studies/phases came together rather than here. 15. Line 36: Suggest adding some rationale on why the development of guiding principles were even needed 16. >Line 48: There seems to be a lot of duplication with results text and Table 5 here. Perhaps just go with the more succinct Table 5. This section also feels like a Discussion section of a paper on the qualitative study and I'm starting to wonder if the authors are trying to do too much in one paper. 17. Page 21, Line 42: What's a COM-B Model? 18. Line 47: As per comments above why were 4 targeted behaviours chosen from the outset? 19. Line 53: You start to mention "expert opinion" like it's been done in this section. Is that what you are referring too and if so it needs more description in earlier mini-studies if that's the case as I missed it? 20. Page 22, >Line 22: May need to explain the different BCM intervention functions and the 18 BCTs in the Methods 21. Line 37: Please perhaps re-draft this sentence as it was a little confusing. 22. Page 23, >Line 27: Figure 1 is very small and difficult to read. However, it is a very useful figure. To my lay way of looking at it would Intervention Processes be better to the left of Intervention Techniques though? Also is there some rationale for the seven processes included in the logic model? It makes sense but is this from the literature and should it be cited or was this the findings of this study? 23. Overall, whilst I'm not an expert in this area, I think the authors are trying to do too much in one paper and they should consider splitting this paper into 2-3 papers, ie qualitative study (and possibly scoping literature review although I'm not sure it added to much I'm sorry to your very nice finding in your qualitative study) and then the guiding
--	--

	principles behaviour analysis/logic model in a separate follow up paper. I feel like I've just read a summary of a thesis in one paper and this one paper perhaps doesn't do the individual mini-studies justice in parts. Discussion  1. Page 24, Line 32: I was going to mention it in your Methods introductory paragraph and its again mentioned here and in the abstract. It might be my lack of experience in this area of development, but I'm not sure how the "theory-, evidence- and person-based approaches ..." fits with what the mini-studies you've just done. Perhaps link each of these three approaches to your mini-studies so its clear to the reader how this fits. 2. Page 25, >Line 8: The authors should cite some of your statements in the discussion and compare your findings to others findings, particularly your qualitative findings to the previous studies you found from the scoping review. At the moment I only count 2 papers cited in the whole discussion which is a little scant for a discussion. 3. Page 26, > Line 8: This section is the first time I realised you hadn't created an intervention and this whole study was really a process to modify the REDUCE program into an electronic REDUCE maintenance program. Perhaps sign post this a little better for the reader in the Background or Methods
--	---

REVIEWER	Jamie Ross UCL, UK
REVIEW RETURNED	28-Nov-2017

GENERAL COMMENTS	This paper is well written, clearly presented and of interest to a wide readership, particularly those interested in the development and evaluation of complex health interventions. The paper presents the planning process for an intervention designed to reduce re-ulceration risk and promote healing for people with diabetes. Theory, evidence and person-based approaches were synthesised to produce a comprehensive plan for intervention development. This paper contributes to the field of intervention development by transparently describing the process by which the intervention was developed, making explicit the intervention contents and hypothesized mechanisms of action. This paper will be valuable to others developing interventions, those wanting to build on research
--

	around the REDUCE intervention and adds to the cumulative science of complex intervention development. Minor comments: Page 5-line 52- Could you provide some more detail on the 'experts', for example, how many experts were there, what were their professional roles/experience, when did they meet, how were they selected? Page 6- line 22- Could you make clear the reasons for selecting to undertake a rapid scoping review as opposed to a systematic review of the literature. Page 9-line 19- How were the topics for the interviews selected? Were they informed by the findings from the scoping review? Could you make clearer whether and how the findings from the scoping review informed the interviews. Page 9-line 43- What were the eligibility criteria for participation in the interviews? Page 24- line 28- Were there any examples of when the different approaches yielded data that weren't complimentary? And how did you resolve this? Were different sources of data given more weight than others when developing the intervention?
--	---

VERSION 1 – AUTHOR RESPONSE

Point by point responses to the reviewer comments are given below.

Editor's comments:

Comment 1: Please separate the methods and results sections.

Following your authorisation, we will present the methods and results individually for each stage, as per original manuscript.

Comment 2: Please ensure that your methods are reported in enough detail so that other researchers would be able to reproduce the study. For example, can the reporting of the scoping review be improved? Why did you choose to conduct your search on Web of Science? What were the dates of coverage? What were the inclusion criteria? etc.

We have added the following information to the scoping review methods section (Page 8, paragraph 1): "This was done to ensure that the initial intervention plan was informed by existing evidence from an early stage. A search was undertaken in Web of Science (covering 1970-2017) to ensure coverage of a range of multidisciplinary journals, easily enabling rapid review."

"It incorporated any published research that included patients who had previously had a diabetic foot ulcer."

"Articles with a biological focus were excluded."

Comment 3: Along with your revised manuscript, please provide a completed copy of the COREQ checklist (<http://www.equator-network.org/reporting-guidelines/coreq/>) for the reporting of the qualitative phase of this study.

We have provided a copy of the COREQ checklist as requested.

Katherine Brown (Reviewer 1) comments:

Comment 4: Use of the abbreviation MI in the abstract - should be explained.

This abbreviation is now written in full as: "maintenance intervention".

Comment 5: Table 4 provides the guiding principles and links these to the key features that addresses them. The reader has not as yet however, been provided with any of the other critical detail that has led to translation of qualitative analysis, literature review and guiding principles into intervention content - feels like we're jumping ahead here. You could consider moving it further down.

The guiding principles (as well as behavioural analysis) take the literature review and qualitative analysis and suggest intervention content and features which are needed based on this evidence. The 5 intervention objectives outlined in Table 4 were derived from the key findings of the qualitative study and scoping review regarding the key characteristics of target users and the key behavioural issues, needs and challenges the intervention must address. The multidisciplinary team decided on the key features based on their ability to address the 5 intervention objectives. The following was added to the methods section of the guiding principles (Page 21) to clarify this process: "The multidisciplinary team decided on the key features based on their ability to address the intervention objectives."

The new text added to paragraph 1 of the 'intervention planning methodology section' on Page 7 also provides an overview of this approach to make it clearer how the evidence informed our intervention planning (through guiding principles and behavioural analysis). We have also added additional subheadings to make the distinction between collecting evidence (review and qualitative study) and the intervention plan to aid the readers understanding.

Comment 6: I think the behavioural analysis table could be presented in the main paper rather than as an appendix.

While we would be happy to do this, BMJ Open permits only 5 tables and figures and we are already at their limit. Therefore, we have prioritised the tables and figures that are essential to the understanding of the main text. We would however be happy to add this as a table in the paper if the Editor would prefer this and would be willing to have 6 tables/figures on this occasion.

Dr Lisa Farndon (Reviewer 2) comments:

Comment 7: In the abstract section MI needs to be written out in full on first mention and again on Page 21, line 49.

This abbreviation is now written in full as: "maintenance intervention".

Comment 8: Background section, page 4, line 17. I disagree that there is a lack of evidence based treatments. There are lots of treatments for diabetic foot ulceration but the main problem is there is a lack of evaluation for the preventative measures on a large scale (regular screening and surveillance,

rapid referral etc.) and whether NG 19 NICE guidance is being followed by all NHS podiatry/diabetic foot services. Please can the authors justify this statement or clarify.

We have clarified our statement as follows: “Although diabetic foot care has been deemed a priority, [2] treatments to prevent ulceration are based largely on expert opinion and small, underpowered, studies [2,8]”

We have also added two supporting references in this statement (a systematic review and the NICE guidelines which reviewed all available evidence on the topic).

Comment 9: Page 6 methods sections. Please provide the date range of which papers were included in the rapid review and justify.

We have added the following information to the scoping review methods section (Page 8, paragraph 1): “A search was undertaken in Web of Science (covering 1970-2017)..”

Comment 10: In the results of the rapid review please can the authors provide information on how the 7 themes were derived, e.g. framework analysis?

We have provided the following clarification in the scoping review methods section (Page 8, paragraph 2): “Using thematic analysis, the key findings were organised into themes relating to the psychosocial and behavioural issues, needs, or challenges to be considered during intervention development.”

Comment 11: Page 9, methods sections. The second paragraph on how the patients were identified should be moved to the opening paragraph of this section to guide the reader on the process.

Suggested change made.

Dr Peter Lazzarini (Reviewer 3) comments:

Comment 12: Page 2, Line 6. Please spell out “REDUCE” or briefly explain what it is. Also please consider this for the title as well.

Line 29: Suggest explaining ‘REDUCE’ a little better. At the moment it is just a little confusing as the program title seems to be an acronym that’s not spelt out or a title for a more general cognitive intervention program that is now being applied to diabetic foot disease or is it just the name of this specific new cognitive intervention for foot disease that’s recently been developed? I’m guessing the later but just clarify please

REDUCE is not an acronym, but a name given to the intervention. In both the title and abstract we have explained that it is a maintenance intervention to reduce re-ulceration risk among patients with a history of diabetic foot ulcers. We have modified the explanation of REDUCE in the background section (Page 5, paragraph 1) as follows to provide clarification:

“‘REDUCE’, a novel complex cognitive behavioural intervention [21], was developed to reduce re-ulceration risk and promote healing by modifying associated psychological and behavioural factors.”

Comment 13: Line 13: As per the main paper I think it would also be useful to explain a little better here how the “mini-studies” in the methods fit together into the larger paper perhaps as escalating phases

Page 5, Line >31: I’m not sure this introductory-type paragraph set the reader up that well for the multiple studies (or phases) that followed. Perhaps towards the end of this intro very clearly state for

the reader the mini-studies/phases that will follow in the methods (even number them) so the reader is not that surprised when they start reading mini-studies/phases as I was initially, eg “This study was undertaken in 4 phases: i) literature review, ii) qualitative interviews, iii) D.”

Page 16, Line 17: As per points above this paragraph I think would be better in the Methods introductory paragraph to explain to the reader how all these mini-studies/phases came together rather than here.

We have added the following explanation of the studies and intervention planning and how these fit together in the intervention planning methodology section on Page 6: “Intervention planning included two phases: collating and analysing evidence; and creating the intervention plan. Phase one includes two elements: a qualitative and quantitative scoping review; and a qualitative interview study. Phase two includes three elements: 1) creating guiding principles; 2) behavioural analysis; and 3) logic modelling.”

We have now structured the paper according to these two phases, with additional subheadings, so it is clear which element fits with each phase and added brief details of the phases in the abstract.

Comment 14: Line 36: As per the main paper I'm not sure you can broadly conclude the “key challenges facing patients” were essentially the four targeted behaviours chosen by the authors prior to the methods. Perhaps just clarify that these challenges emanated from the four targeted behaviours only

Apologies for the confusion, we have modified this sentence to clarify the challenges were relating to the 4 target behaviours: “Key challenges to the interventions’ target behaviours included.”

Comment 15: Line 45: What is MI? I can't remember it being mentioned in the main paper

This abbreviation is now written in full as: “maintenance intervention”.

Comment 16: Line 49. Are the “intervention processes outlined” here a finding of your study or just what the literature suggested should be broadly covered in a logic model framework? Possibly consider removing this sentence.

We have amended this sentence to now read: “The behavioural analysis identified the following processes hypothesised to facilitate long-term behaviour maintenance including; increasing patients’ skills...”

Comment 17: Page 3, Line 15: This point is great, but I didn't get this in the main paper until the last sentence of the Discussion, perhaps add something like this in the Background of the main paper somewhere.

We have added the following in the last paragraph on Page 5 to emphasise this point sooner: “In keeping with recent NICE research priorities, this will be done through behaviour change and emotional management.”

Comment 18: Page 4, Line 17: Suggest cite this statement re: lack of treatments to prevent ulceration as there are papers in References that will support this important statement in your rationale.

We have clarified this statement as follows: “Although diabetic foot care has been deemed a priority,[2] treatments to prevent ulceration are based largely on expert opinion and small, underpowered, studies [2,8]”

We have also added two supporting references (a systematic review and the NICE guidelines which reviewed all available evidence on the topic).

Comment 19: Line 24: Perhaps be more specific with “such factors” that NICE have recommended, ie what were those factors as this would add nicely to your rationale.

NICE do not specify what psychosocial and behavioural factors have been implicated. We have added the following to paragraph 2 on Page 4 to provide more information on the existing evidence-base for these behaviours:

“Evidence suggests an association between longer delays in help seeking and increased ulcer severity, highlighting the importance of regular foot-checking and rapid self-referral.[14] Although physical activity is generally encouraged in diabetes to promote glycaemic control and reduce cardiovascular risk, there is a common assumption that greater physical activity may increase ulceration risk in people at risk of DFUs. However, research suggests that moderate, regular activity may decrease risk, or at worst, be unrelated to risk.[15,16] Emotional management may also play a role. Following a diabetic foot ulcer (DFU), people may experience difficult emotions, including depression, blame, and guilt.[17] Depression has been associated with greater ulcer incidence and recurrence, and a slower rate of ulcer healing.[18-20]”

We have also added the following information about factors addressed in the original REDUCE programme to Page 5, Paragraph 1: “These factors include; non-adherence to recommended foot care procedures (e.g. foot checking), delayed help-seeking for changes in foot health, low or irregular levels of physical activity, and difficulties in managing negative emotions.”

Comment 20: Line 42: Did REDUCE produce improved (patient) outcomes or was it just a pilot feasibility study? Please briefly add a sentence or two on what its outcomes were that you are now intending to build on.

We have added the following detail on the original REDUCE feasibility study published by Vedhara et al to Page 5, Paragraph 2: “A feasibility study found REDUCE to be acceptable and feasible for patients and preliminary descriptive findings suggested that patients experienced changes in many of the psychological and behavioural factors targeted by the intervention. [22]”

Comment 21: Line 44: I think I know what you mean by “digital maintenance” but perhaps clarify this a little better, ie you are essentially trying to build on the REDUCE program and also turn it into a web-based electronic design?

Line 49: Possibly add a sentence to Segway a little better between these paragraphs to say REDUCE maintenance intervention is what you are going to do in this study. At the moment it sounds like you are still discussing someone else’s program and not the one you are just about to describe in your paper.

Also how does this plan build on the original REDUCE program? We don’t seem to have any details on the original REDUCE program but it seems like the reader should know what it is.

Page 4, Consider explaining the original REDUCE program a little better, how this development procedure builds on the original program and that you are intending to then develop a further intervention after this paper from what you have learnt from this paper.

Page 26, > Line 8: This section is the first time I realised you hadn’t created an intervention and this whole study was really a process to modify the REDUCE program into an electronic REDUCE maintenance program. Perhaps sign post this a little better for the reader in the Background or Methods

We have added the following to Page 5, Paragraph 2 to clarify that the current intervention will replace the original maintenance phase of REDUCE and provide a Segway to the paragraph about the REDUCE maintenance intervention: "This paper describes the planning process for an intervention that will replace the face-to-face maintenance sessions of the original intervention."

We have added the following information about the factors addressed in the original programme to Page 5, Paragraph 1: "These factors include; non-adherence to recommended foot care procedures (e.g. foot checking), delayed help-seeking for changes in foot health, low or irregular levels of physical activity, and difficulties in managing negative emotions."

We have also added the following to Page 5, Paragraph 2 to signpost readers to where they can find further detail about the programme: "A full description of the intervention can be found in Vedhara et al. [22]"

Comment 22: Its not my area of expertise; however, I'm not familiar with essentially reporting 4 or 5 small studies consecutively such as this in one large paper like a mini-thesis. I'm wondering if it would be easier for the reader to conceptualise using phases (as you seemed to have used four escalating phases) and report those phases under one methods section and then report the results of those phases in one results section instead of all together in one big section?

Page 6, I'd suggest 'signposting' the 4-5 mini-studies/phases a little better early in the Methods and then perhaps consider writing the paper as a typical research paper, i.e. firstly a methods section outlining the methods of the 4-5 phases of development and then a results section reporting the results of these 4-5 phases; rather than combining all into one big section as it is at present. /

Separate the methods and results sections.

Methods/Results section. Overall, whilst I'm not an expert in this area, I think the authors are trying to do too much in one paper and they should consider splitting this paper into 2-3 papers, ie qualitative study (and possibly scoping literature review although I'm not sure it added to much I'm sorry to your very nice finding in your qualitative study) and then the guiding principles behaviour analysis/logic model in a separate follow up paper. I feel like I've just read a summary of a thesis in one paper and this one paper perhaps doesn't do the individual mini-studies justice in parts.

Reviewers report: However, I couldn't help but feel the authors were also trying to fit too much into one paper. It felt like I was reading a summary of an entire thesis in one paper that perhaps didn't do the individual mini-studies in the paper justice. I would suggest the authors consider splitting this paper into 2 or 3 papers as the qualitative study could easily have been a paper in its own right and I thought it was a very nice study that got a little lost with everything else going on. If the authors do consider to keep going with one paper as is, may I suggest you consider explaining the original REDUCE program a little better, how this development procedure builds on the original program and that you are intending to then develop a further intervention after this paper from what you have learnt from this paper. Also I'd suggest 'signposting' the 4-5 mini-studies/phases a little better early in the Methods and then perhaps consider writing the paper as a typical research paper, i.e. firstly a methods section outlining the methods of the 4-5 phases of development and then a results section reporting the results of these 4-5 phases; rather than combining all into one big section as it is at present.

We have discussed this with the editor and have agreed to present the methods and results individually for each stage, as per original manuscript. We have some concerns about whether your suggestion of reporting the phases under one methods and one results section could make the article harder to follow. This is because the reader would have to remember the methods for each part of the intervention planning for some time before reading the corresponding results for that stage of intervention planning. We believe that this would be challenging and might reduce coherence and flow of the article because of the 5 stages of intervention planning involved. We previously published a similar article planning another intervention in a different field and had originally structured this to

have one large methods and results section, as you suggested. However, the feedback from reviewers was that this was very difficult for the reader as they couldn't remember the methods for each part of the results because of the multiple stages involved in intervention planning and so needed to keep moving backwards between methods and results which was made the article difficult to follow (article: (<https://implementationscience.biomedcentral.com/articles/10.1186/s13012-017-0553-4>). We hope our new restructuring also highlights that there are not actually 5 mini studies, there are two studies (a scoping review and a qualitative study) and then an intervention planning phase which has 3 components (guiding principles, behavioural analysis and logic model). We hope this new structure and our additional explanation of the overview of this process in the methods section helps to make this clearer.

Comment 23: Line 48: I'm not sure the "Additional barriers and facilitators D were identified through consultation with our multidisciplinary project team, including clinical and intervention development experts." was described later on like the other mini-studies/phases were. Consider adding something on how you obtained this "expert opinion".

We have added the following to Page 7, Paragraph 2: "Expert opinion was gained through iterative consultation at regular teleconferences and feedback on drafts of the intervention plan."

Comment 24: Line 22: Were you looking for health professional's views as well as patients? The results seem to indicate patients only. I would have thought patients only would suffice though, but remove health professionals if that was the case.

Our review included studies exploring the views of both patients and health professionals. Table 1 has been modified to clarify which of the results came from patients or health professionals.

Comment 25: Line 27: It would be useful to provide some rationale why the particular four behaviours were chosen as I assumed this was going to come from the rapid search exploring experiences of managing DFU or DFU history. However, the search only used terms for the four behaviours. Possibly also add this explanation to where they were mentioned in the Background as well.

Page 12, Line 5: I'm still not sure why any of the 4 targeted behaviours were chosen? I can assume why a few of them were, however, physical activity for example seems an odd one out as limited solid evidence on what type and intensity of physical activity promotes ulcer prevention and also where the "pedometer" discussion came from? Again it would be useful to have some rationale around why/how these 4 targeted behaviours were originally chosen. If it was from the original REDUCE study [14] then this rationale needs to be in the background, but if it was part of this study then it should go in the methods somewhere perhaps?

Line 47: As per comments above why were 4 targeted behaviours chosen from the outset?

These behaviours were chosen based on their inclusion in the original REDUCE intervention. We have added in a discussion of the evidence for the association between these four behaviours and DFU outcomes on Page 4, paragraph 2: "Evidence suggests an association between longer delays in help seeking and increased ulcer severity, highlighting the importance of regular foot-checking and rapid self-referral.[14] Although physical activity is generally encouraged in diabetes to promote glycaemic control and reduce cardiovascular risk, there is a common assumption that greater physical activity may increase ulceration risk in people at risk of DFUs. However, research suggests that moderate, regular activity may decrease risk, or at worst, be unrelated to risk.[15,16] Emotional management may also play a role. Following a diabetic foot ulcer (DFU), people may experience difficult emotions, including depression, blame, and guilt.[17] Depression has been associated with greater ulcer incidence and recurrence, and a slower rate of ulcer healing.[18-20]"

We have also added the following information about factors addressed in the original REDUCE programme to Page 5, paragraph 1: "These factors include; non-adherence to recommended foot

care procedures (e.g. foot checking), delayed help-seeking for changes in foot health, low or irregular levels of physical activity, and difficulties in managing negative emotions.]”

The aim of our literature review was to review evidence examining the behavioural and psychosocial needs, issues, and challenges of people who have had DFUs. It did not aim to identify which behaviours were important for reducing re-ulceration risk. Ideas for intervention features (e.g. pedometers) were chosen based on the multidisciplinary team’s knowledge of the evidence for the acceptability and effectiveness of these features for changing the target behaviours.

Comment 26: Line 48: I think the authors need to state the seven articles and themes were those applicable to the 4 behaviours they specifically targeted rather than relating this to all patient beliefs around foot care etc.

The themes identified were broader than just the four behaviours investigated (e.g. concerns over using digital interventions, difficult emotions following a DFU). We have added the following to Page 8, paragraph 1 for clarification: “Findings regarding beliefs around foot care were excluded if they were only relevant to foot care behaviours not targeted in the REDUCE maintenance intervention (e.g. barriers to adherence to prescription footwear)”

The following sentence on Page 8, paragraph 2 was also modified to clarify this: “The review identified seven articles and highlighted six themes relating to people’s beliefs around DFUs and the target behaviours”

Comment 27: Page 8, Table 1: In the results it said 7 themes were elicited from the rapid lit review but there are only 6 themes in the table

This typo has been corrected to say 6 themes.

Comment 28: Page 9, Line 8: I think this mini-study needs some brief explanation of what the intervention was that was presented to the patient participants. Further, this mini-study doesn’t seem to link to the first mini-study that well and seems to beg the question why was study 1 performed when it was always the intention to use a program already developed? Perhaps clarify how these 2 mini-studies fit together a little better.

Studies 1 and 2 were carried out in parallel and offered two different complementary insights for intervention development. We have added an explanation of how these two studies complement each other in the last paragraph of Page 7: “These two studies are both person- and evidence-based approaches as they aim to develop an in-depth understanding of the patients’ perspective (person-based approach), while identifying, summarising, and incorporating the evidence-base on the barriers and facilitators to the target behaviours (evidence-based approach).”

Participants were not presented with an already developed intervention. They were merely presented with some initial intervention ideas (see Appendix 1). The prompt cards outlined the four behaviours, why these behaviours are important and potential intervention components that could be included in the intervention (e.g. use of pedometers, email reminders). We have modified the following sentence in the purpose section on page 11 to make it clear that participants were only presented initial ideas for the intervention, rather than an already developed intervention: “To explore the acceptability and feasibility of initial ideas regarding the content and delivery of the maintenance intervention...”

Comment 29: Line >17: Suggest some of the methods are more findings (results) of the methods and to put them in results, e.g. “66 patients expressed interest D “ and Table 2 should probably go under results also.

We have kept all of the information regarding the participants in the methods section as this is consistent with guidance on reporting qualitative research (e.g. the COREQ checklist; see Comment 3).

Comment 30: Page 11, Table 3: There are some really nice and unique results in this table and in the following few pages from your qualitative study here. This study could be a paper on its own. Perhaps move the table down under some of the first results text though as I didn't realise there were more results summarised in text

We have moved Table 3 to after the regular foot-checking section on Page 16.

Comment 31: Page 13, Line 35: What were the planned intervention features? It would be useful to have described them before here I think

We have provided a description of the key intervention features in the last paragraph of Page 11: "Interviews explored participants' views of the target behaviours and potential intervention features, including foot checking reminders, facilities for note-taking, personalised advice about when to self-refer, advice on pacing physical activity, goal setting, provision of free pedometers, and emotional management techniques." These are also presented on the prompt cards within our interview schedule (see appendix 1).

Comment 32: Line 36: Suggest adding some rationale on why the development of guiding principles were even needed

Guiding principles are consistent with the person-based approach we have chosen to take for this research. Further explanation of the need for PBA is provided in the intervention planning methodology section.

We have added the following rationale in the purpose section on Page 21 to clarify this: "In line with the person-based approach, [26] brief guiding principles are developed and consulted throughout intervention development to ensure that the intervention is underpinned by a coherent focus."

Comment 33: >Line 48: There seems to be a lot of duplication with results text and Table 5 here. Perhaps just go with the more succinct Table 5. This section also feels like a Discussion section of a paper on the qualitative study and I'm starting to wonder if the authors are trying to do too much in one paper.

We feel the explanation in the main text is needed to demonstrate how the findings from the scoping review and qualitative interviews were used to create each guiding principle, this is in line with our Person-Based Approach which calls for this approach of showing how the findings were translated into intervention design objectives.

Comment 34: Page 21, Line 42: What's a COM-B Model?

As explained in the text in Page 25, paragraph 2, the COM-B model is a model of behaviour change that "argues that behaviour is influenced by an individual's Capability, Opportunity, and Motivation to change behaviour". A reference is provided to allow readers to read up on this model in more depth if need be.

Comment 35: Page 21, Line 53: You start to mention "expert opinion" like it's been done in this section. Is that what you are referring too and if so it needs more description in earlier mini-studies if that's the case as I missed it?

More information about the expert consultation has now been added to the intervention planning methodology section of the methods and results. We have added more detail about who was in the multidisciplinary project team and the expert consultation process on Page 7, paragraph 2: “This team included one diabetologist, two diabetes specialist podiatrists, one diabetes specialist nurse, one cognitive behavioural psychotherapist, five health psychologists, and one research psychologist.”

“Expert opinion was gained through iterative consultation at regular teleconferences and feedback on drafts of the intervention plan.”

Comment 36: Page 22, >Line 22: May need to explain the different BCM intervention functions and the 18 BCTs in the Methods

We did not feel it was possible to provide a comprehensive explanation of the 6 BCW functions and 18 BCTs within the allocated word count. References providing in-depth descriptions of each of these components are provided for readers who require more detail.

Comment 37: Line 37: Please perhaps re-draft this sentence as it was a little confusing.

This sentence has been modified as follows: “Although participants would have liked additional health professional support, the support participants wanted was more clinical in nature (e.g. advice about foot health or when to self-refer). As such support would be provided in the website/booklet, this form of health professional support was deemed superfluous.”

Comment 38: Page 23, >Line 27: Figure 1 is very small and difficult to read. However, it is a very useful figure. To my lay way of looking at it would Intervention Processes be better to the left of Intervention Techniques though? Also is there some rationale for the seven processes included in the logic model? It makes sense but is this from the literature and should it be cited or was this the findings of this study?

The logic model outlines the logical progression of relationships between variables (i.e. the potential causal / chronological order of relationships). The techniques specified are hypothesised to affect the specified processes, hence why techniques precede processes in the model. We have amended the sentence in the intervention techniques and processes section on Page 27 as follows to clarify where these intervention processes came from: “These are the psychosocial factors that need to be modified for the intervention’s target behaviours to change and were identified through the behavioural analysis.”

Comment 39: Page 24, Line 32: I was going to mention it in your Methods introductory paragraph and its again mentioned here and in the abstract. It might be my lack of experience in this area of development, but I’m not sure how the “theory-, evidence- and person-based approachesD” fits with what the mini-studies you’ve just done. Perhaps link each of these three approaches to your mini-studies so its clear to the reader how this fits.

We have modified the following sentences in the last paragraph of Page 6; Page 7, paragraph 3; and Page 29, paragraph 1 to clarify which studies apply to which approach: “These two studies [review and qualitative study] are both person- and evidence-based approaches as they aim to develop an in-depth understanding of the patients’ perspective (person-based approach), while identifying, summarising, and incorporating the evidence-base on the barriers and facilitators to the target behaviours (evidence-based approach).”

“In line with a person-based approach, [26] all sources of evidence...”

“In line with person- and evidence-based approaches, our scoping review and qualitative study deepened our understanding”

Comment 40: Page 25, >Line 8: The authors should cite some of your statements in the discussion and compare your findings to others findings, particularly your qualitative findings to the previous studies you found from the scoping review. At the moment I only count 2 papers cited in the whole discussion which is a little scant for a discussion.

We have now added more references to the discussion and expanded our discussion of our findings from the different studies:

“For example, the scoping review highlighted that patients experience difficult emotions following DFUs, , however, the qualitative interviews suggested that this was only relevant for some patients, suggesting that this content should be made optional.”

“Some of which had been highlighted in the literature (e.g. lack of knowledge regarding what to look for when foot checking [17,30]) and some which had received little prior attention (e.g. lack of knowledge about when to self-refer).”

“Our qualitative study updated prior research published over a decade ago that highlighted concerns regarding limited computer access and poor computer skills among people at risk of DFUs.[32]”

“...will allow other researchers to easily understand how this methodology could be applied to different intervention contexts and facilitate comparison between different interventions. [12,23-25]”

Other comments: Page 6, Line 15: Would it be better to use “review” than “collate”?

Page 5, Line 17: Would “describes” be a better term than “presents” to give the reader the impression that this will be a descriptive paper?

All suggested edits have been implemented.

Jamie Ross (Reviewer 4) comments:

Comment 41: Page 5-line 52- Could you provide some more detail on the ‘experts’, for example, how many experts were there, what were their professional roles/experience, when did they meet, how were they selected?

We have added more detail on Page 7, paragraph 2 about who was in the multidisciplinary project team and the expert consultation process: “This team included one diabetologist, two diabetes specialist podiatrists, one diabetes specialist nurse, one cognitive behavioural psychotherapist, five health psychologists, and one research psychologist.”

“Expert opinion was gained through iterative consultation at regular teleconferences and feedback on drafts of the intervention plan.”

Comment 42: Page 6- line 22- Could you make clear the reasons for selecting to undertake a rapid scoping review as opposed to a systematic review of the literature.

We have added the following rationale to Page 8, paragraph 1: “This was done to ensure that the initial intervention plan was informed by existing evidence from an early stage.”

Comment 43: Page 9-line 19- How were the topics for the interviews selected? Were they informed by the findings from the scoping review? Could you make clearer whether and how the findings from the scoping review informed the interviews.

The scoping review and qualitative study were carried out in parallel. Therefore, the scoping review did not inform the qualitative study. Rather they provided two complementary insights into the perspectives of this target group. The topics for the interviews were selected based on the intervention components that we wanted to explore perceptions of and based on our interest in which modality of intervention delivery might be optimal.

Comment 44: Page 9-line 43- What were the eligibility criteria for participation in the interviews?

The following sentence on Page 11, paragraph 2 now reads: “A total of 250 adult (aged 18+ years) patients with diabetes who had previously had a DFU were contacted by letter by their local NHS podiatry service. Participants were excluded if they had a DFU in the previous two weeks.”

Comment 45: Page 24- line 28- Were there any examples of when the different approaches yielded data that weren't complimentary? And how did you resolve this? Were different sources of data given more weight than others when developing the intervention?

None of the results from the qualitative study and review were conflicting, but some were complementary. We have added an example of how these different approaches yielded complementary insights, and how we refined our conclusions when results from the studies differed on Page 28, last paragraph: “For example, the scoping review highlighted that patients experience difficult emotions following DFUs,[17,31-34] however, the qualitative interviews suggested that this was only relevant for some patients, suggesting that this content should be made optional.”

We gave the qualitative study and review findings equal weight. We have added the following clarification to Page 7, paragraph 1: “The findings of these two studies were given equal weight when creating the intervention plan.”

VERSION 2 – REVIEW

REVIEWER	Dr Lisa Farndon Sheffield Teaching Hospitals NHS Foundation Trust
REVIEW RETURNED	15-Jan-2018
GENERAL COMMENTS	Thank you to the authors for making the changes suggested by the reviewers. I feel this paper is a lot clearer now and sufficiently sets out the results of the study in a logical and clear manner.
REVIEWER	Katherine E Brown Coventry University, UK
REVIEW RETURNED	30-Jan-2018
GENERAL COMMENTS	This is an extremely clearly presented manuscript. I have no major concerns with it in its current form and would consider it to be a valuable piece for publication in BMJ Open. It would be useful however to understand a bit more about the way in which the consultation with experts and professionals specifically contributed to development, as this was the only area that was lighter on methodological detail and outcomes. Reference to a 'research

	psychologist' is also a bit vague, they must have some specialism or expertise which could be referenced.
--	---

VERSION 2 – AUTHOR RESPONSE

Editorial Request

Comment 1: “Please change the 'Design' sub-heading in the abstract to 'Methods'.”

Response: We have made the requested change.

Reviewer 1 (Katherine E Brown) Requests

Comment 2: “It would be useful however to understand a bit more about the way in which the consultation with experts and professionals specifically contributed to development, as this was the only area that was lighter on methodological detail and outcomes.”

Response: We have re-written the relevant paragraph on page 7 to make it clearer regarding how the expert consultation contributed to the intervention development (see below).

“We also consulted with experts in diabetic foot ulcers, behaviour change, and intervention development who belonged to our multidisciplinary project team using regular teleconferences to discuss and gain feedback on drafts of the intervention plan. This team included one diabetologist, two diabetes specialist podiatrists, one diabetes specialist nurse, one cognitive behavioural psychotherapist, five health psychologists, and one research psychologist specialising in health. From this, additional barriers and facilitators were identified, and suggestions or refinements to intervention content were made.”

Comment 3: “Reference to a 'research psychologist' is also a bit vague, they must have some specialism or expertise which could be referenced.”

Response: We have added on page 7 that the research psychologist specialises in health; ‘health psychologist’ is a protected title so we have avoided its use here as the team member is not a chartered health psychologist (see below).

“...one research psychologist specialising in health”.

VERSION 3 – REVIEW

REVIEWER	Katherine E Brown Coventry University, UK
REVIEW RETURNED	05-Mar-2018

GENERAL COMMENTS	Further minor concerns have now been addressed
--